# Severe COVID-19 Outcomes in Five Latin American Countries in the Postvaccination Era

**DOI:** 10.3390/v16071025

**Published:** 2024-06-26

**Authors:** Guilherme Silva Julian, Júlia Spinardi, Melissa Diaz-Puentes, Diana Buitrago, Ida Caterina García, Moe H. Kyaw

**Affiliations:** 1Evidence Generation Medical Affairs, Pfizer Inc., São Paulo 04717-904, Brazil; 2Vaccines Medical Affairs, Pfizer Inc., São Paulo 04717-904, Brazil; julia.spinardi@pfizer.com; 3Faculdade de Ciências Médicas da Santa Casa de São Paulo, São Paulo 01224-001, Brazil; 4Real World Insights (RWI), IQVIA, Bogotá 110110, Colombia; dianacamila.buitrago@iqvia.com; 5Real World Insights (RWI), IQVIA, Ciudad de México C.P. 03810, Mexico; idacaterina.garcia@iqvia.com; 6Vaccines Clinical Epidemiologist Emerging Markets, Pfizer Inc., Collegeville, PA 19426-3982, USA; moe.kyaw@pfizer.com

**Keywords:** coronavirus, COVID-19, Latin America, risk factors, hospitalization rate, fatal outcomes

## Abstract

We conducted a multicountry retrospective study using data from COVID-19 national surveillance databases to analyze clinical profiles, hospitalization rates, intensive care unit (ICU) admissions, utilization of ventilatory support, and mortality rates in five Latin American countries in the context of COVID-19 vaccination implementation. We analyzed the sociodemographic characteristics, comorbidities, clinical outcomes, and vaccination status of laboratory-confirmed COVID-19 cases from January 2021 to December 2022. We calculated the yearly and quarterly hospitalization rates per 1000 confirmed COVID-19 cases and ICU admissions, use of mechanical ventilators, and mortality rates per 1000 hospitalized cases, with their corresponding 95% confidence interval (CI) of 38,209,397 confirmed COVID-19 cases. Rates of hospitalization, ICU admission, ventilatory support, and death were higher among males than among females (30.6 vs. 25, 275.9 vs. 218.8, 156.4 vs. 118.6, and 388.4 vs. 363.1 per 1000, respectively); higher in 2021 than in 2022 (51.6 vs. 20.2, 471.4 vs. 75.5, 230.1 vs. 46.7, and 307.9 vs. 230.3 per 1000, respectively); and higher in the >50 age group (range: 4.3–16.3, 35.5–149.5, 20.1–83.2, and 315–462.9, per 1000) than the <50 age group (range: 0.8–5.7, 3.0–49.3, 2.1–39.3, and 7.8–217.7 per 1000). Hypertension and diabetes mellitus were the most common comorbidities in Mexico and Colombia. Prevention and treatment strategies for these case profiles could bring benefits from a public health perspective.

## 1. Introduction

The coronavirus disease 2019 (COVID-19) pandemic remains a major but preventable global health challenge, and Latin America (LATAM) has been one of the most affected regions. By October 2023, according to the World Health Organization (WHO), 6,7139,997 confirmed cases and 2,618,286 deaths had been reported across Argentina, Chile, Colombia, Mexico, and Brazil [1,2]. Moreover, a number of Severe acute respiratory syndrome coronavirus 2 (SARS-CoV-2) variants of interest and concern have circulated in LATAM and the variants Gamma (P.1), Lambda (C.37), and Mu (B.1.621) were first identified within the region [3].

COVID-19 vaccines have proven to be essential to controlling the pandemic by providing protection against symptomatic and severe disease worldwide [4,5]. In LATAM, multiple COVID-19 vaccine platforms were used, achieving high vaccination coverage among adults for the primary schedule. Subsequently, the messenger ribonucleic acid (mRNA) platform became preferred for administering booster doses. The timeline for vaccine introduction varied across countries, but in general, elderly individuals and adults with comorbidities were prioritized in vaccination programs [6]. Treatment strategies were also implemented. Recently, a pan-American guideline indicated the use of different antivirals according to patient characteristics and clinical presentation, with a strong recommendation for the use of nirmatrelvir/ritonavir in outpatients with mild COVID-19 [7]. Clear documentation of antiviral access and availability in the region is not available.

There is a large number of publications about the rates of severe outcomes in patients with COVID-19 conducted in high-income and developed countries. However, studies on this topic within LATAM are scarce and often limited to local data and small sample sizes. COVID-19 outcomes are expected to be influenced by the public policies implemented in each country, which also reflect existing vulnerabilities. Therefore, understanding the epidemiology and outcomes of patients with COVID-19 in countries within LATAM would provide key insights to be better prepared for future emerging diseases. Moreover, as the disease transitions to an endemic stage, identifying individuals at risk of severe outcomes in high vaccination coverage settings becomes crucial for guiding future strategies, potentially combining prevention and treatment. Thus, we aimed to analyze clinical profiles, hospitalization rates, intensive care unit (ICU) admissions, utilization of ventilatory support, and mortality rates in Argentina, Brazil, Colombia, Chile, and Mexico in the context of COVID-19 vaccination implementation.

## 2. Materials and Methods

### 2.1. Study Design

We conducted a multicounty retrospective study using COVID-19 surveillance databases from Brazil, Colombia, Mexico, Chile, and Argentina. We included all confirmed cases based on laboratory tests (e.g., Reverse Transcription Polymerase Chain Reaction or antigenic tests from January 2021 to December 2022.

### 2.2. Participants and Procedures

For Brazil, we used databases available on the openDATASUS platform, which is managed by the Brazilian Ministry of Health (MoH) [8]. We used the eSUS-Notifica SG database for cases with mild to moderate acute respiratory syndrome (Sistema de Informação de Vigilância Sentinela de Sindrome Gripal—SG database). For cases with severe acute respiratory syndrome, we accessed the SIVEP-Flu database (Sistema de Informação de Vigilância Epidemiológica da Gripe—Influenza Epidemiological Surveillance Information System). We obtained data on the total population count from the Instituto Brasileiro de Geografia e Estatística.

In Mexico, COVID-19 data are centrally managed through the Epidemiological Surveillance System for Viral Respiratory Diseases. This system receives reports from a network of 475 Viral Respiratory Disease Monitoring Health Units (USMER) from both the public sector, primarily the Mexican Social Security Institute (Instituto Mexicano del Seguro Social—IMSS), the Institute for Social Security and Services for State Workers (Instituto de Seguridad y Servicios Sociales de los Trabajadores del Estado—ISSSTE), the Secretary of National Defense (Secretaría de la Defensa Nacional—SEDENA), and the navy (Secretaría de Marina—SEMAR), as well as the private health sector. Cases are categorized as ambulatory or hospitalized based on their clinical diagnosis at admission. The database used includes information on sociodemographics, comorbidities, hospitalization, intubation, and the use of ICUs; however, it does not include information about vaccination [9]. Deaths recorded within this database are in the inpatient setting. We obtained the total population count from the National Institute of Statistics and Geography of Mexico (Instituto Nacional de Estadística y Geografía —INEGI) database.

For Colombia, we used the Integrated Social Protection Information System (Sistema Integrado de Información de la Protección Social—SISPRO), a government-owned database comprising information on the health sector, covering approximately 99% of the Colombian population in 2021 (Appendix A). Its COVID-19 module contains case-level information on confirmed COVID-19 cases (date of diagnosis, comorbidities, sociodemographic characteristics, and needed hospitalization) reported by health institutions through the application SEGCOVID-19. Additionally, we obtained mortality data from the RUAF (Regísto Único de Afiliados) module and vaccination status from the PAIweb application. We retrieved information about cases who needed ventilatory support from another SISPRO module, the Individual Registry of Health Services Provision module (Registro Individual de Prestación de Servicios de Salud—RIPS). We extracted ventilatory support data from this database filtering by the procedure codes defined in Appendix A and the International Classification of Diseases, Tenth Revision (ICD-10) diagnosis code U071. Due to a lack of linkage between the RIPS database and the COVID-19 module, we could not determine the comorbidities or vaccination status of cases requiring ventilatory support in this country [10]. We extracted the total population count from estimations provided by the DANE (Departamento Administrativo Nacional de Estadística).

In Argentina, data from the Ministry of Health (MoH) are available individually for each confirmed COVID-19 case from January 2021 to June 2022. The database includes sociodemographic variables (age, sex, and place of residence) and clinical outcomes (hospitalization, ICU admission, ventilatory support, and mortality) [11]. We extracted the total population count from the INDEC (Instituto Nacional de Estadística y Censos) estimations.

In the case of Chile, aggregated data, presented as total counts of cases over specific periods, are available in different data products from the MoH [12]. We analyzed the information between week 40 of 2021 and week 51 of 2022, as this study period was available in all data products. The database does not provide information regarding the use of ventilatory support. As age groups differed from the ones defined for the other countries, we reported them separately. We extracted the total population from the INE (Instituto Nacional de Estadísticas) database.

Across all countries, we analyzed sociodemographic variables such as age, sex, and ethnicity, as well as comorbidities and vaccination status (when available). The outcomes included hospitalization in the general ward, ICU admission, use of ventilatory support defined as mechanical ventilation, and mortality.

### 2.3. Statistical Analysis

We conducted descriptive analyses to summarize the sociodemographic characteristics, comorbidities, and vaccination status of confirmed COVID-19 cases. For categorical variables, we estimated frequencies and percentages. We calculated yearly and quarterly hospitalization rates per 1000 confirmed COVID-19 cases and ICU admissions, use of mechanical ventilators, and mortality rates per 1000 hospitalized cases, with their corresponding 95% confidence interval (CI). We calculated rates based on the number of inhabitants per year in each country, considering age and sex, utilizing data from each country’s demographic and statistical institute. Additionally, we analyzed incidence rates according to the predominant SARS-CoV-2 variants quarterly by country, as described in Appendix A. All statistical analyses were conducted using Python version 3.8.10 [13].

## 3. Results

A total of 46,176,672 COVID-19-confirmed cases were retrieved across countries’ surveillance databases. After excluding cases with duplicate or missing information (Appendix A), our study included 38,209,397 confirmed COVID-19 cases, of which 45.7% were from Brazil, 14.2% were from Mexico, 12.2% were from Colombia, 18.7% were from Argentina, and 8.5% were from Chile, in the period 2021–2022. 

In the overall population, the proportion of female COVID-19 cases (55.1%) exceeds that of males, and most cases were between 18 and 49 years old (62.3%) (Table 1). In Brazil, 37.4% of all confirmed cases were reported among vaccinated individuals, whereas in Colombia, the corresponding percentage was higher, at 74.4%. In the aggregate data from all five countries, 2,122,629 cases required hospitalization (5.6%), and 769,941 died (2.3%) between 2021 and 2022. The percentage of cases who needed ventilatory support was 2.9% (all countries with available data), and 1.3% of the cases needed ICU admission (all countries with available data). 

The age and sex distributions in each country were similar to the overall study population. Fatal outcomes occurred in an average of 2.3% of the confirmed cases in all countries, ranging from 0.2% in Chile to 2.9% in Mexico. The percentage of cases that needed hospitalization ranged from 1.6% in Chile to 7.4% in Brazil. The proportion of cases who needed ventilatory support and ICU admission was also higher in Brazil (5.3% and 2.5%, respectively) than in other countries (below 1%).

### 3.1. Analysis of Severe Outcomes in COVID-19-Confirmed Cases

#### 3.1.1. Hospitalizations

Incidence rates were greater for male COVID-19 cases (30.6 per 1000 confirmed cases) than for female cases (25.0 per 1000 confirmed cases), and this trend was observed across all countries for the entire study period (Table 2). The age group with the greatest hospitalization rates was the 50–64 age group in all countries (21.4 for Brazil, 19.5 for Mexico, 11.3 for Colombia, and 4.7 for Argentina), except for Chile, where higher rates were observed in cases who were in the 71–80 age group (3.2 per 1000 confirmed cases). Hospitalization rates decreased from 51.6 in 2021 to 20.2 in 2022 overall, and in all countries individually except for Mexico (Table 3 and Appendix A). In Mexico and Colombia, the prevailing comorbidities among hospitalized cases consistently included hypertension (35.3% and 32.3%, respectively) and diabetes mellitus (DM) (29.6% and 13.9%, respectively). Decompensated chronic respiratory diseases (35.5%), carriers of chromosomal diseases or immunologically fragile states (35.4%), and hematologic disease (35.1%) were the most reported comorbidities in Brazil (Appendix A).

#### 3.1.2. Ventilator Use

The need for ventilatory support was more frequent in male COVID-19 cases than in female cases in all countries in which data were available (282.9 vs. 236.2 per 1000 hospitalized cases) (Table 3). Rates of ventilatory support were highest among hospitalized cases aged 50–64 years (135.8 per 1000 hospitalized cases). Ventilatory use rates decreased substantially over time, from 441.5 in 2021 to 114.9 per 1000 hospitalized cases in 2022. Brazil accounted for 96.1% of the ventilatory support cases identified in this study.

#### 3.1.3. ICU Admissions

ICU admission rates were higher among males than among females in all countries where information was available (148.4 vs. 117.7 per 1000 hospitalized cases) (Table 3). Rates of ICU admissions were highest among 50–64-year-old cases in Brazil, Mexico, and Argentina (81.0, 20.5, and 77.6 per 1000 hospitalized cases, respectively); in Chile, ICU admission dates were highest among cases 61–70 years old (17.7 per 1000 hospitalized cases). Compared with 2021, the rate of ICU admissions in 2022 decreased in Brazil and Argentina (from 242.8 to 63.3 and from 210.0 to 31.9 per 1000 hospitalized cases, respectively). However, this trend was not observed in Mexico.

#### 3.1.4. Deaths

Mortality rates were higher among males than among females (388.4 vs. 363.1 per 1000 hospitalized cases) and increased with age in all countries (Table 4). Overall, mortality decreased over time in all countries except Mexico, where rates remained constant throughout the whole study period (Appendix A).

#### 3.1.5. Comorbidities

In Mexico and Colombia, the most frequent comorbidities among COVID-19 cases requiring ventilatory support (Appendix A), ICU admission (Appendix A), or who died (Appendix A) were hypertension, DM, and obesity, whereas decompensated chronic respiratory diseases and carrier chromosomal disease were the most common comorbidities reported in Brazil.

### 3.2. Vaccination

Across all age groups, the proportion of unvaccinated individuals was highest among deceased cases, followed by hospitalized and non-hospitalized cases (Figure 1). In Brazil, the proportion of unvaccinated cases was 70.5% among those requiring hospitalization (Appendix A), 71.3% among cases needing ICU admission (Appendix A), 70.4% among cases requiring ventilatory support (Appendix A), and 71.1% among cases with fatal outcomes (Appendix A). In Colombia, 42.6% of cases requiring hospital admission and 68.9% of cases with a fatal outcome were unvaccinated (Appendix A).

Among non-hospitalized cases in Brazil and Colombia, those aged 65–74 years had the highest proportion of complete vaccination schedules (54% and 82%, respectively) (Figure 1). The proportion of completely vaccinated individuals among hospitalized and deceased cases was highest among those aged 85+ years: 50% and 44%, respectively, in Brazil and 53% and 42%, respectively, in Colombia.

### 3.3. Incidence Rates by Variant

In 2021, the 20J Gamma V3 variant predominated in Brazil (in Q1, Q2, and Q3), Mexico (in Q3 and Q4), and Argentina (in Q2 and Q3), coinciding with elevated rates of hospitalization (23.0, 15.7, and 9.0 per 1000 confirmed cases, respectively), ventilatory support (232.5, 19.8, and 79.9 per 1000 confirmed cases, respectively), and ICU admission (106.7 and 13.1 per 1000 confirmed cases, respectively) (Table 3 and Table 4 and Appendix A). The same year, the 21H Mu variant was predominant in Colombia from Q1 to Q3, correlating with the highest hospitalization rate (17.9 per 1000 confirmed cases) in Q2.

In 2022, Omicron was present in all countries. The 22B Omicron variant was the most frequent in Brazil (in Q3 and Q4) and Mexico (in Q3 and Q4), coinciding with the lowest reported rates of hospitalization (1.8 and 2.9 per 1000 confirmed cases, respectively), ventilatory support (13.7 and 1.9 per 1000 confirmed cases, respectively), and ICU admission (7.3 and 3.2 per 1000 confirmed cases, respectively) in Q4. Colombia’s most frequent variant was 22A Omicron in Q3 and Q4, coinciding with the lowest hospitalization rate (0.6 per 1000 confirmed cases) in Q4. In Argentina, multiple Omicron sublineages were present throughout the year (21K, 21L, 22B, and 22E), with the lowest rates of hospitalization (0.2 per 1000 confirmed cases), ventilatory support (0.6 per 1000 confirmed cases), and ICU admission (1.6 per 1000 confirmed cases) observed in Q2 (Table 2 and Table 3, and Appendix A).

## 4. Discussion

This study analyzed the clinical profiles, rates of hospitalization, ICU admission, ventilatory support utilization, and mortality of COVID-19 cases in five LATAM countries following the implementation of national COVID-19 vaccination programs. We observed a decrease in hospitalization rates from 2021 to 2022, but our findings reinforce that severe outcomes of COVID-19 persist with a considerable burden in the region.

Previously published studies from both pre- and postvaccination periods [14,15] in Mexico reported that most of the COVID-19 patients who needed hospitalization, ICU care, and ventilatory support were males rather than females. This tendency was also observed in all countries in our study, consistent with a study in Colombia that included COVID-19 patients in 2020–2021 and reported a higher frequency of cases among males for all outcomes [16]. Similarly, a Brazilian retrospective study that included 74,991 fully vaccinated COVID-19 patients from February 2021 to January 2022 also reported a higher proportion of male cases compared with female cases [17]. In addition, a retrospective cohort study conducted among COVID-19 patients in Chile showed that male sex was strongly associated with a poor prognosis of COVID-19 [18].

Sex has been reported as a risk factor for COVID-19 outcomes outside LATAM. A nationwide, registry-based study in Sweden confirmed that male sex was a risk factor for COVID-19 hospitalization [19]. Additionally, a multicenter retrospective cohort study in the USA using the Rush University System concluded that male sex was independently associated with death, hospitalization, ICU admission, and the need for vasopressors or endotracheal intubation [20].

In all countries, we found that the rates of hospitalization, ventilatory support, and ICU admission increased with age, especially among cases aged 50 or older. This finding aligns with previous studies indicating that people over 60 years are at greater risk of serious complications from COVID-19, increasing the probability of requiring hospitalization in both LATAM countries (Mexico and Chile) and European countries (France) [14,21,22]. Similarly, in the US, severe outcomes such as hospitalization and death are substantially more common among elderly persons than in other age groups [23]. All these data highlight that efforts to increase the use of booster COVID-19 vaccines are particularly important to reduce the burden of severe COVID-19 in the elderly population [24].

There are limited data on the impact of comorbidities on COVID-19 clinical outcomes in LATAM countries. In this study, the main comorbidities among cases with severe clinical outcomes were hypertension, DM, and obesity. This is consistent with previous research among Mexican patients, demonstrating that DM, hypertension, and obesity are significantly associated with severe COVID-19 [14,15,25]. Likewise, studies carried out in Colombia showed that during the Mu wave, hypertension was the main comorbidity in patients with COVID-19, and hypertension along with DM were associated with severe conditions [16]. Further additional studies to determine the independent risk factors for severe COVID-19 outcomes will aid in prioritizing vaccination or treatment guidelines for selected high-risk patient groups. While this study could not provide data on comorbidities within the Chilean population, overall findings agree with those from previous studies from Chile that found that hypertension and DM were the most common comorbidities among patients throughout all COVID-19 waves [26].

A potential correlation between hospitalization rates and the predominant SARS-CoV-2 variant in each country was evident, particularly during the first quarter of 2022. This period coincided with the emergence of the 21K Omicron variant, which led to a notable surge in hospitalizations in Mexico, Argentina, and Colombia. During this same period, we observed a decline in mortality rates in Colombia and Argentina, aligning with previous findings reported by Calonico and Álvarez [27,28]. This decrease is likely attributed to a better understanding of the disease as well as increased adherence to global evidence-based guidelines [27]. This observation highlights the importance of monitoring and studying the impact of different variants on healthcare systems and the importance of ongoing research into the dynamics of SARS-CoV-2 variants and their impacts on public health.

This study addresses a critical knowledge gap in the understanding of COVID-19 outcomes, particularly in countries with varying vaccination rates. We identified that the proportion of unvaccinated individuals was highest among deceased cases, followed by hospitalized and non-hospitalized cases. However, in both Colombia and Brazil, a considerable number of cases with complete primary vaccination schedules still required hospitalization. This suggests that vaccination alone may not be sufficient to prevent all severe cases [29]. Efforts should be made to complete vaccination schedules and promote booster vaccination in LATAM, especially among people older than 50 years and those with comorbidities such as hypertension and diabetes.

This study had some limitations. Regarding the change in compulsory notification, for example, in Argentina starting in June 2022, information on patients who were treated in an ambulatory setting was mandatorily reported only if patients were aged over 50 years, had risk factors, or belonged to special populations [30]. As the transition continues from a pandemic to an endemic state, countries have reduced safety measures, creating a lower perception of risk in the general population [31] and reducing healthcare-seeking behaviors associated with COVID-19. To mitigate these limitations, we chose hospitalization, ICU admission, ventilatory support, and death as outcomes, as these are reported to all countries’ surveillance systems. Due to the characteristics of the study data sources, it was not possible to analyze treatment patterns such as antiviral drugs in outpatients as a tool to reduce hospitalization and severe outcomes, as well as inpatient care. Finally, information about the date of vaccination in the data sources in Brazil and Colombia was incomplete in most cases, which limited the possibility of analyzing the vaccination status according to the time from vaccination to the occurrence of each COVID-19 outcome. This issue highlights the importance of improving data accuracy and completeness in national surveillance databases to enable more robust analyses of vaccine implementation in LATAM.

Although this study has inherent limitations due to its retrospective nature and data variability among countries, it benefits from using nationwide surveillance systems and real-world data. Additionally, the study benefits from a large sample size and incorporates data from multiple countries within LATAM, offering a comparative analysis that can inform public health strategies and identify areas for data quality improvement in infectious disease surveillance.

Our findings emphasize the ongoing importance of preventive and treatment measures for the early identification and appropriate treatment of at-risk patients. Population-level surveillance systems are critical to monitoring the impact of COVID-19 and implementing timely effective prevention and treatment guidelines in the postvaccine era in LATAM countries.

## Figures and Tables

**Figure 1 viruses-16-01025-f001:**
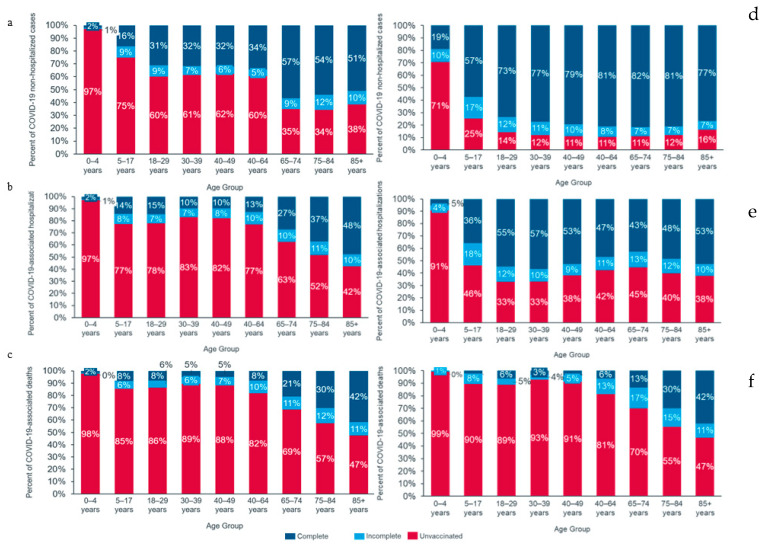
Vaccination status by age group in Brazil and Colombia for non-hospitalized, hospitalized, and deceased COVID-19 patients. Figures (**a**–**c**) correspond to Brazil, and figures (**d**–**f**) correspond to Colombia. Figures (**a**,**d**) correspond to non-hospitalized cases, (**b**,**e**) to COVID-19-associated hospitalizations, and (**c**,**f**) to COVID-19-associated deaths. Bars in red represent the proportion of unvaccinated cases among each age group. Bars in light blue correspond to the proportion of cases with an incomplete COVID-19 primary vaccination schedule and dark blue to the proportion of cases with a complete COVID-19 primary vaccination schedule among each age group. The proportion of unvaccinated cases was the highest among the deceased, followed by hospitalized cases. The proportion of cases with complete vaccination schedules was highest among older age groups for all displayed outcomes.

**Table 1 viruses-16-01025-t001:** Characteristics of the study population (confirmed COVID-19 cases per country).

	Brazil	Mexico	Colombia	Argentina	Chile	Total
	(*n* = 17,751,803)	(*n* = 5,427,997)	(*n* = 4,644,148)	(*n* = 7,131,436)	(*n* = 3,254,013)	(*n* = 38,209,397)
Percent of participants	45.7%	14.2%	12.2%	18.7%	8.5%	100.0%
Confirmed COVID-19 rate per 1000 individuals	82.6	28.2	156.1	154.2	-	-
Surveillance period	January 2021–December 2022	January 2021–December 2022	January 2021–December 2022	January 2021– June 2022	October 2021–December 2022	-
Sex, *n* (%)						
Female	9,834,763 (55.4)	2,953,304 (54.4)	2,440,147 (52.5)	3,721,818 (52.2)	2,119,577 (54.2)	2,1069,609 (55.1)
Male	7,916,338 (44.6)	2,474,693 (45.6)	2,195,856 (47.3)	3,386,340 (47.5)	1,788,553 (45.8)	17,761,780 (46.5)
Missing	702 (0.0)	0 (0)	0 (0)	23,278 (0.3)	0 (0)	23,980 (0.1)
Age group, *n* (%)						
0–4 years	339,244 (1.9)	60,385 (1.1)	95,500 (2.1)	51,908 (0.7)	-	547,037 (1.6) a
5–17 years	1,213,990 (6.8)	332,247 (6.1)	334,631 (7.2)	435,339 (6.1)	-	2,316,207 (6.6) a
18–29 years	3,459,429 (19.5)	1,345,869 (24.8)	1,044,569 (22.5)	1,747,598 (24.5)	-	7,597,465 (21.7) a
30–39 years	3,635,229 (20.5)	1,269,871 (23.4)	1,008,790 (21.7)	1,654,993 (23.2)	-	7,568,883 (21.7) a
40–49 years	3,419,927 (19.3)	1,047,737 (19.3)	789,303 (17)	1,366,862 (19.2)	-	6,623,829 (18.9) a
50–64 years	1,140,200 (6.4)	956,156 (17.6)	872,746 (18.8)	1,208,934 (17)	-	4,178,036 (12) a
65–74 years	239,098 (1.3)	257,761 (4.7)	291,481 (6.3)	409,769 (5.7)	-	1,198,109 (3.4) a
75–84 years	173,128 (1.0)	117,057 (2.2)	145,466 (3.1)	184,836 (2.6)	-	620,487 (1.8) a
85+ years	99,387 (0.6)	40,912 (0.8)	61,605 (1.3)	69,845 (1)	-	271,749 (0.8) a
Missing	4,032,171 (22.7)	2 (0)	57 (0)	1352 (0)	-	4,033,582 (11.5) a
3–5 years	-	-	-	-	56,541 (1.4)	56,541 (1.4) b
6–11 years	-	-	-	-	174,800 (4.5)	174,800 (4.5) b
12–20 years	-	-	-	-	346,671 (8.9)	346,671 (8.9) b
21–30 years	-	-	-	-	617,741 (15.8)	617,741 (15.8) b
31–40 years	-	-	-	-	652,629 (16.7)	652,629 (16.7) b
41–50 years	-	-	-	-	495,282 (12.7)	495,282 (12.7) b
51–60 years	-	-	-	-	416,448 (10.7)	416,448 (10.7) b
61–70 years	-	-	-	-	278,912 (7.1)	278,912 (7.1) b
71–80 years	-	-	-	-	142,332 (3.6)	142,332 (3.6) b
80+ years	-	-	-	-	72,657 (1.9)	72,657 (1.9) b
Missing	-	-	-	-	654,117 (16.7)	654,117 (16.7) b
Patients who needed hospitalization, *N* (%)	1,307,618 (7.4)	375,146 (6.9)	195,321 (4.2)	128,373 (1.8)	116,171 (1.6)	2,122,629 (5.6)
Hospitalization rate per 1000 Confirmed COVID-19 cases	73.7	69.1	42.1	18.0	35.7	-
Patients who needed ventilatory support, *N* (%)	938,850 (5.3)	33,612 (0.6)	1515 (0)	18,728 (0.3)	-	992,705 (2.9) a
Ventilatory support rate per 1000 hospitalized COVID-19 cases	718.0	89.6	7.8	145.9	-	-
Patients who needed ICU admission, *N* (%)	443,359 (2.5)	25,572 (0.5)	-	31,236 (0.4)	7868 (0.1)	508,035 (1.3) c
ICU admission rate per 1000 hospitalized COVID-19 cases	339.1	68.2	-	243.3	67.7	-
Patients who had a fatal outcome, N (%)	424,606 (2.4)	158,298 (2.9)	94,354 (2)	79,615 (1.1)	13,068 (0.2)	769,941 (2.3)
Mortality rate of COVID-19 patients per 1000 hospitalized cases	324.7	422.0	483.1	620.2	112.5	-

a, Percentages in the total number of cases excluding Chile; b, percentages in the total number of cases in Chile; c, percentages in the total number of cases excluding Colombia.

**Table 2 viruses-16-01025-t002:** Hospitalization rates per 1000 confirmed COVID-19 cases.

	Brazil		Mexico		Colombia		Argentina		Chile		Total	
	(*n* = 1,307,618)		(*n* = 375,146)		(*n* = 195,321)		(*n* = 128,373)		(*n* = 116,171)		(*n* = 2,122,629)	
	Rate	95% CI	Rate	95% CI	Rate	95% CI	Rate	95% CI	Rate	95% CI	Rate	95% CI
Percent of participants	61.1%		17.7%		9.2%		6.0%		5.5%		100.0%	
Surveillance period	January 2021–December 2022		January 2021–December 2022		January 2021–December 2022		January 2021–June 2022		October 2021–December 2022			
Sex												
Female	33.2	33.1–33.3	30.8	30.7–30.9	18	17.9–18.1	7.9	7.8–8	17.5	17.4–17.6	25	24.9–25.1
Male	40.5	40.4–40.5	38.4	38.2–38.6	24	23.9–24.1	9.9	9.8–10	18.2	18.1–18.3	30.6	30.5–30.7
Age group												
0–4 years	1.2	1.2–1.3	1.4	1.4–1.4	1.8	1.8–1.8	0.3	0.3–0.3	-	-	1.1	1.1–1.1
5–17 years	0.7	0.7–0.7	1.5	1.5–1.5	0.7	0.7–0.7	0.4	0.4–0.4	-	-	0.8	0.7–0.8
18–29 years	3.1	3.1–3.2	4.4	4.3–4.5	1.9	1.9–1.9	0.8	0.8–0.8	-	-	2.7	2.7–2.7
30–39 years	7.8	7.7–7.8	6.7	6.6–6.8	3.1	3–3.2	1.3	1.3–1.3	-	-	5.7	5.7–5.7
40–49 years	11.6	11.6–11.7	9.4	9.3–9.5	4.7	4.6–4.8	2.2	2.2–2.2	-	-	8.4	8.4–8.4
50–64 years	21.4	21.3–21.4	19.5	19.4–19.6	11.3	11.2–11.4	4.7	4.6–4.8	-	-	16.3	16.3–16.4
65–74 years	13	12.9–13.0	13.1	13–13.2	8.1	8–8.2	3.5	3.5–3.5	-	-	10.4	10.4–10.4
75–84 years	9.4	9.4–9.5	9.1	9–9.2	6.7	6.6–6.8	2.9	2.9–2.9	-	-	7.7	7.7–7.7
85+ years	5.4	5.4–5.4	4	3.9–4.1	3.8	3.7–3.9	1.9	1.9–1.9	-	-	4.3	4.2–4.3
3–5 years	-	-	-	-	-	-	-	-	0.1	0.1–0.1	0.1	0.1–0.1
6–11 years	-	-	-	-	-	-	-	-	0.2	0.2–0.2	0.2	0.2–0.2
12–20 years	-	-	-	-	-	-	-	-	0.5	0.5–0.5	0.5	0.5–0.5
21–30 years	-	-	-	-	-	-	-	-	1.2	1.2–1.2	1.2	1.2–1.2
31–40 years	-	-	-	-	-	-	-	-	1.4	1.4–1.4	1.4	1.4–1.4
41–50 years	-	-	-	-	-	-	-	-	1.3	1.3–1.3	1.3	1.3–1.3
51–60 years	-	-	-	-	-	-	-	-	2.0	2–2	2.0	2–2
61–70 years	-	-	-	-	-	-	-	-	2.8	2.7–2.9	2.8	2.7–2.9
71–80 years	-	-	-	-	-	-	-	-	3.2	3.1–3.3	3.2	3.1–3.3
80+ years	-	-	-	-	-	-	-	-	3.1	3–3.2	3.1	3–3.2
Period												
2021Q1	23	23.0–23.1	5.5	5.4–5.6	9.5	9.4–9.6	3.0	3–3	-	-	14.4	14.4–14.4
2021Q2	20.8	20.7–20.8	1.8	1.8–1.8	17.9	17.8–18	9.0	8.9–9.1	-	-	15.1	15.1–15.1
2021Q3	10.3	10.3–10.4	15.7	15.6–15.8	5.2	5.1–5.3	2.7	2.7–2.7	-	-	8.9	8.9–8.9
2021Q4	7.9	7.9–8.0	9.5	9.4–9.6	2.0	2–2	0.5	0.5–0.5	-	-	5.9	5.9–5.9
2022Q1	5.1	5.1–5.2	16.6	16.5–16.7	5.0	4.9–5.1	2.6	2.6–2.6	-	-	6.4	6.4–6.4
2022Q2	2.6	2.6–2.7	6.4	6.3–6.5	0.8	0.8–0.8	0.2	0.2–0.2	-	-	2.5	2.5–2.5
2022Q3	1.8	1.7–1.8	10.0	9.9–10.1	1.2	1.2–1.2	-	-	-	-	2.6	2.6–2.6
2022Q4	2.1	2.1–2.1	2.9	2.9–2.9	0.6	0.6–0.6	-	-	-	-	1.6	1.6–1.6
Year												
2021	62.1	62.0–62.2	32.5	32.3–32.7	34.6	34.4–34.8	15.1	15–15.2	14.8	14.7–14.9	51.6	51.5–51.7
2022	11.6	11.5–11.6	35.8	35.6–36	7.5	7.4–7.6	2.8	2.8–2.8	20.9	20.7–21.1	20.2	20.2–20.2

Rates among the total number of confirmed COVID-19 cases in countries with available information.

**Table 3 viruses-16-01025-t003:** Ventilatory support and ICU admission rates per 1000 hospitalized COVID-19 cases.

	Brazil		Mexico		Colombia		Argentina		Chile		Total	
	Rate	95% CI	Rate	95% CI	Rate	95% CI	Rate	95% CI	Rate	95% CI	Rate	95% CI
Ventilatory Support	(*n* = 938,850)		(*n* = 33,612)		(*n* = 1515)		(*n* = 18,728)		-		(*n* = 992,705)	
Percent of participants	94.6%		3.4%		0.2%		1.9%		0.00%		100.00%	
Surveillance period	1/2021–12/2022		1/2021–12/2022		1/2021–12/2022		1/2021–6/2022		-		-	
Sex												
Female	319.8	319.0–320.6	34.9	34.3–35.5	3.3	3–3.6	54.5	53.2–55.8	-	-	218.8 a	218.2–219.4
Male	398.1	397.3–399.0	54.7	54–55.4	4.5	4.2–4.8	90.7	89.1–92.3	-	-	275.9 a	275.2–276.6
Age group												
0–4 years	7.8	7.7–8.0	1.1	1–1.2	0.1	0.1–0.1	0.6	0.5–0.7	-	-	5.4 a	5.3–5.5
5–17 years	4.3	4.2–4.4	0.9	0.8–1	0	0–0	0.9	0.7–1.1	-	-	3 a	2.9–3.1
18–29 years	26	25.7–26.3	3.2	3–3.4	0.2	0.1–0.3	3.2	2.9–3.5	-	-	17.8 a	17.6–18
30–39 years	72.7	72.3–73.2	7.1	6.8–7.4	0.5	0.4–0.6	8	7.5–8.5	-	-	49.3 a	49–49.6
40–49 years	113.8	113.3–114.3	12.7	12.3–13.1	1.2	1–1.4	18.3	17.6–19	-	-	77.8 a	77.4–78.2
50–64 years	215.6	214.9–216.3	29.2	28.7–29.7	2.8	2.6–3	50.7	49.5–51.9	-	-	149.5 a	149–150
65–74 years	131	130.4–131.6	19.7	19.3–20.1	1.7	1.5–1.9	37.3	36.2–38.4	-	-	91.6 a	91.2–92
75–84 years	93.6	93.1–94.1	11.9	11.6–12.2	1.1	1–1.2	21	20.2–21.8	-	-	64.7 a	64.3–65.1
85+ years	52.8	52.5–53.2	3.8	3.6–4	0.2	0.1–0.3	5.8	5.4–6.2	-	-	35.5 a	35.2–35.8
Period												
2021Q1	232.5	231.8–233.3	9.5	9.2–9.8	-	-	23.3	22.5–24.1	-	-	171.5 b	170.9–172.1
2021Q2	211.6	210.9–212.3	2.7	2.5–2.9	-	-	79.9	78.4–81.4	-	-	159 b	158.4–159.6
2021Q3	103.6	103.1–104.1	19.8	19.3–20.3	-	-	23	22.2–23.8	-	-	80.5 b	80.1–80.9
2021Q4	78.9	78.5–79.4	15.3	14.9–15.7	-	-	3.7	3.4–4	-	-	60.4 b	60–60.8
2022Q1	41.7	41.3–42.0	17.3	16.9–17.7	-	-	15.5	14.8–16.2	-	-	34.7 b	34.4–35
2022Q2	20.3	20.0–20.5	8.1	7.8–8.4	-	-	0.6	0.5–0.7	-	-	16.4 b	16.2–16.6
2022Q3	13.7	13.5–13.9	13.5	13.1–13.9	-	-	-		-	-	12.7 b	12.5–12.9
2022Q4	15.7	15.5–16.0	1.9	1.8–2	-	-	-		-	-	11.8 b	11.6–12
Year												
2021	626.6	625.8–627.4	47.3	46.6–48	-	-	129.2	127.2–131.2	-	-	471.4 b	470.4–472.4
2022	91.4	90.9–91.9	40.7	40.1–41.3	-	-	16	15.3–16.7	-	-	75.5 b	75.1–75.9
ICU admission	(*n* = 443,359)		(*n* = 25,572)		-		(*n* = 31,236)		(*n* = 7868)		(*n* = 508,035)	
Percent of participants	87.3%		5.0 %		0.00%		6.1%		1.5%		100.00%	
Surveillance period	1/2021–12/2022		1/2021–12/2022		1/2021–12/2022		1/2021–6/2022		10/2021–12/2022		-	
Sex												
Female	147.7	147.1–148.3	27.5	27–28	-	-	95.1	93.4–96.8	-	-	119.1 a	118.6–119.6
Male	191.3	190.6–192.0	40.7	40.1–41.3	-	-	146.8	144.7–148.9	-	-	157 a	156.4–157.6
Age group												00
0–4 years	4.2	4.1–4.3	1.5	1.4–1.6	-	-	1.5	1.3–1.7	-	-	3.4 a	3.3–3.5
5–17 years	2.4	2.3–2.5	1	0.9–1.1	-	-	2.1	1.8–2.4	-	-	2.1 a	2–2.2
18–29 years	10.9	10.7–11.1	3.4	3.2–3.6	-	-	6.5	6.1–6.9	-	-	9 a	8.9–9.1
30–39 years	29.7	29.4–30.0	6.5	6.2–6.8	-	-	14.4	13.7–15.1	-	-	23.8 a	23.6–24
40–49 years	48.6	48.3–49.0	9.9	9.6–10.2	-	-	29.6	28.7–30.5	-	-	39.3 a	39–39.6
50–64 years	101.2	100.7–101.7	20.5	20–21	-	-	77.6	76.1–79.1	-	-	82.8 a	82.4–83.2
65–74 years	68.5	68.0–68.9	13	12.6–13.4	-	-	58.9	57.6–60.2	-	-	56.3 a	56–56.6
75–84 years	48.4	48.1–48.8	8.8	8.5–9.1	-	-	37.8	36.7–38.9	-	-	39.5 a	39.2–39.8
85+ years	25	24.7–25.2	3.7	3.5–3.9	-	-	15	14.2–15.6	-	-	19.9 a	19.7–20.1
3–5 years	-	-	-	-	-	-	-	-	0.4	0.3–0.5	0.4 b	0.3–0.5
6–11 years	-	-	-	-	-	-	-	-	0.5	0.4–0.6	0.5 b	0.4–0.6
12–20 years	-	-	-	-	-	-	-	-	1	0.8–1.2	1 b	0.8–1.2
21–30 years	-	-	-	-	-	-	-	-	2.4	2.1–2.7	2.4 b	2.1–2.7
31–40 years	-	-	-	-	-	-	-	-	4.7	4.3–5.1	4.7 b	4.3–5.1
41–50 years	-	-	-	-	-	-	-	-	6.5	6.0–7.0	6.5 b	6.0–7.0
51–60 years	-	-	-	-	-	-	-	-	12.1	11.5–12.7	12.1 b	11.5–12.7
61–70 years	-	-	-	-	-	-	-	-	17.7	16.9–18.5	17.7 b	16.9–18.5
71–80 years	-	-	-	-	-	-	-	-	15.8	15.1–16.5	15.8 b	15.1–16.5
80+ years	-	-	-	-	-	-	-	-	6.6	6.1–7.1	6.6 b	6.1–7.1
Period												
2021Q1	106.7	106.2–107.3	4.5	4.3–4.7	-	-	41.4	40.3–42.5	-	-	80.9 a	80.5–81.3
2021Q2	94.2	93.7–94.7	2	1.9–2.1	-	-	123.6	121.7–125.5	-	-	77.1 a	76.7–77.5
2021Q3	50.6	50.2–50.9	13.1	12.7–13.5	-	-	39.2	38.1–40.3	-	-	42 a	41.7–42.3
2021Q4	38.1	37.8–38.5	6.9	6.6–7.2	-	-	7.1	6.6–7.6	-	-	29.5 a	29.2–29.8
2022Q1	22.5	22.3–22.8	15	14.6–15.4	-	-	30.4	29.4–31.4	-	-	21.5 a	21.3–21.7
2022Q2	10.9	10.8–11.1	9.2	8.9–9.5	-	-	1.6	1.4–1.8	-	-	9.9 a	9.8–10
2022Q3	7.3	7.1–7.4	13.2	12.8–13.6	-	-	-	-	-	-	8 a	7.9–8.1
2022Q4	8.7	8.5–8.9	3.2	3–3.4	-	-	-	-	-	-	6.9 a	6.8–7
Year												
2021	289.6	288.8–290.4	26.4	25.9–26.9	-	-	210	207.5–212.5	-	-	229.4 a	228.7–230.1
2022	49.5	49.1–49.8	40.7	40.1–41.3	-	-	31.9	30.9–32.9	-	-	46.4 a	46.1–46.7

a, Rates among the total number of cases excluding Chile and Colombia; b, rates among the total number of cases in Chile.

**Table 4 viruses-16-01025-t004:** Mortality rates of COVID-19 cases per 1000 hospitalized cases.

	Brazil		Mexico d		Colombia		Argentina		Chile b		Total	
	(*n* = 424,606)		(*n* = 158,298)		(*n* = 94,354)		(*n* = 79,615)		(*n* = 13,068)		(*n* = 769,941)	
	Rate	95% CI	Rate	95% CI	Rate	95% CI	Rate	95% CI	Rate	95% CI	Rate	95% CI
Percent of participants	55.1%		20.6%		12.3%		10.3%		1.7%		100.0%	
Surveillance period	1/2021–12/2022		1/2021–12/2022		1/2021–12/2022		1/2021–6/2022		10/2021–12/2022			
Sex												
Female	322.6	321.4–323.8	382.9	380–385.9	450.4	445.8–454.9	597.2	590.8–603.6	-	-	363.1 a	361.9–364.3
Male	326.4	325.3–327.5	453.3	450.4–456.1	508.4	504.2–512.6	637.4	631.5–643.2	-	-	388.4 a	387.2–389.6
Age group												
0–4 years	45.8	43.1–48.6	54.9	49.6–60.1	14.7	12.1–17.3	43.3	33.8–52.8	-	-	7.8 a	7.4–8.2
5–17 years	70.8	66.3–75.4	55.8	50.6–61	39.2	32.5–45.8	55.0	46.2–63.8	-	-	20.6 a	19.6–21.6
18–29 years	119.4	116.7–122.1	129.7	125.1–134.3	136.8	129.1–144.5	117.6	108.7–126.5	-	-	95.6 a	93.9–97.3
30–39 years	153.2	151.3–155.1	230.3	225.4–235.3	236.8	228.9–244.7	206.9	197.8–216.1	-	-	152.5 a	150.9–154.1
40–49 years	207.3	205.5–209.0	333.5	328.5–338.5	347.9	340–355.7	336.4	327.3–345.5	-	-	217.7 a	216.1–219.3
50–64 years	310.1	308.6–311.5	452.4	448.4–456.5	464.7	458.9–470.5	539.2	531.3–547	-	-	315 a	313.6–316.4
65–74 years	434.3	432.3–436.3	550.1	544.7–555.6	592.7	584.9–600.5	794.3	783.3–805.3	-	-	406.8 a	404.9–408.7
75–84 years	490.9	488.5–493.3	583.2	576.5–590	672.5	663.3–681.6	922.5	909.5–935.6	-	-	434.9 a	432.7–437.1
85+ years	545.3	542.1–548.4	596.7	586.4–606.9	811.3	798.1–824.5	1063.3	1045.7–1080.9	-	-	462.9 a	4599–4659
3–5 years	-	-	-	-	-	-	-	-	12.0	1.5–22.6	12 c	1.5–22.5
6–11 years	-	-	-	-	-	-	-	-	12.9	4–21.9	12.9 c	4–21.8
12–20 years	-	-	-	-	-	-	-	-	17.1	10.9–23.3	17.1 c	10.9–23.3
21–30 years	-	-	-	-	-	-	-	-	19.9	15.5–24.4	19.9 c	15.5–24.3
31–40 years	-	-	-	-	-	-	-	-	35.3	29.8–40.8	35.3 c	29.8–40.8
41–50 years	-	-	-	-	-	-	-	-	89.6	80.4–98.7	89.6 c	80.4–98.8
51–60 years	-	-	-	-	-	-	-	-	149.5	140–159	149.5 c	140–159
61–70 years	-	-	-	-	-	-	-	-	205.7	196.3–215.1	205.7 c	196.3–215.1
71–80 years	-	-	-	-	-	-	-	-	318.8	307.9–329.7	318.8 c	307.9–329.7
80+ years	-	-	-	-	-	-	-	-	624.8	609.4–640.2	624.8 c	609.4–640.2
Period												
2021Q1	360.2	358.7–361.7	32.4	31.9–33	444.0	437.8–450.3	635.2	624.5–646	-	-	193.9 a	193–194.8
2021Q2	305.6	304.1–307.1	8.8	8.5–9.1	568.8	563.7–573.9	661.4	655.1–667.7	-	-	209.8 a	208.9–210.7
2021Q3	316.2	314.0–318.3	29.4	29.1–29.7	498.0	489.1–507	581.1	570.4–591.8	-	-	76.1 a	75.7–76.5
2021Q4	330.7	328.2–333.2	47.1	46.5–47.7	394.2	381.3–407	472.6	449.6–495.6	-	-	100.4 a	99.7–101.1
2022Q1	325.7	322.6–328.7	31.5	31.2–31.8	400.0	391.8–408.2	558.9	548.3–569.6	-	-	63.7 a	63.3–64.1
2022Q2	260.7	256.7–264.7	34.3	33.7–34.8	198.0	183.2–212.8	205.4	180.7–230.1	-	-	50.5 a	49.9–51.1
2022Q3	274.8	269.9–279.8	32.9	32.5–33.3	239.6	226.9–252.4	-	-	-	-	41.6 a	41.2–42
2022Q4	263.7	259.2–268.1	28.4	27.7–29.1	224.5	206.4–242.5	-	-	-	-	53.9 a	53.1–54.7
Year												
2021	330.8	329.9–331.7	434.8	431.7–437.9	514.0	510.5–517.5	635.1	630.3–639.9	-	-	307.9 a	307.1–308.7
2022	292	290.1–294.0	409.0	406.1–411.8	340.3	334.2–346.4	536.3	526.1–546.4	-	-	230.3 a	229.2–231.4

a, Rates among the total number of cases excluding Chile; b, data from week 40 of 2021 to week 51 of 2022; c, rates among the total number of cases in Chile; d, mortality rates among hospitalized COVID-19 cases.

## Data Availability

The original contributions presented in the study are included in the article and Appendix A, further inquiries can be directed to the corresponding author.

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
