# Peer review of "Severe COVID-19 Outcomes in Five Latin American Countries in the Postvaccination Era"

_viruses, 2024, doi:10.3390/v16071025_

Round 1

Reviewer 1 Report

Comments and Suggestions for Authors

1.  These authors have used national surveillance databases to analyze clinical profiles, hospitalization rates, intensive care unit admissions, the use of ventilatory support, and mortality rates in 5 Latin American countries in the time period following the introduction of COVID-19 vaccination.  This study included 38,852,831 confirmed COVID-19 cases, and they used this information to calculate the rates per 1000 cases.
2.  They were able to confirm that the cases were higher in men than women, higher in older age groups, and higher in patients with hypertension and diabetes as comorbidities.  Tables 1 through 4 report this information in detail.
3.  The figure illustrates the effect of vaccination on infection nonhospitalized cases, hospitalized cases, and cases with mortality in various age groups.
4.  The discussion analyzes the effect of age, comorbidity, and viral variants on these various outcomes.  Although the reader can easily reach conclusions with this information, it would be useful if the authors provided their own recommendations as to which individuals warrant more effort for referral for vaccination and completion of vaccination series.

Author Response

Response to Reviewer 1 comments

Thank you very much for taking the time to review this manuscript. Please find the detailed responses below and the corresponding revisions highlighted in the re-submitted files.

General comments

  1. These authors have used national surveillance databases to analyze clinical profiles, hospitalization rates, intensive care unit admissions, the use of ventilatory support, and mortality rates in 5 Latin American countries in the time period following the introduction of COVID-19 vaccination. This study included 38,852,831 confirmed COVID-19 cases, and they used this information to calculate the rates per 1000 cases.
  2. They were able to confirm that the cases were higher in men than women, higher in older age groups, and higher in patients with hypertension and diabetes as comorbidities. Tables 1 through 4 report this information in detail.
  3. The figure illustrates the effect of vaccination on infection nonhospitalized cases, hospitalized cases, and cases with mortality in various age groups.

Response to the general comments:  Thank you for your thorough review and insightful comments on our manuscript. We appreciate your recognition of the key findings of our study.

Point-by-point response to Comments and Suggestions for Authors

Comment:  The discussion analyzes the effect of age, comorbidity, and viral variants on these various outcomes. Although the reader can easily reach conclusions with this information, it would be useful if the authors provided their own recommendations as to which individuals warrant more effort for referral for vaccination and completion of vaccination series.

Response:  Thanks for this suggestion We agree with your comment and have included our recommendations regarding this topic in the discussion section, page 13, lines 303 to 305.

Reviewer 2 Report

Comments and Suggestions for Authors

This manuscript provided big data on COVID-19 outcomes among five Latin Americans. The writing and outcomes were impressive and exciting. Overall parameters were essential to the manuscript. This manuscript could be of interest to the audience. However, some points need to be addressed to improve the quality of the manuscript.

Major concerns.

1. COVID-19 vaccination status: The COVID-19 vaccination is an important factor in substantially reducing the chance of disease severity.

I suggest adding further information on COVID-19 vaccination;

1) Doses - you may be using a categorial data such as
- 0, 1, 2, 3, 4, 5+
- never, partial-, full-, booster

2) Last dose interval to infection - suggest analysing further data to reveal insight information. As you know, recent vaccination within 3 months could protect the host from severe conditions than late.
- <14 days, 14-35 days (1 month), 2 months, 3 months, over 3 months.

2. Did this study have smoking status data?

Author Response

Response to Reviewer 2 comments

Thank you very much for taking the time to review this manuscript. Please find the detailed responses below and the corresponding revisions highlighted in the re-submitted files.

General comments

This manuscript provided big data on COVID-19 outcomes among five Latin Americans. The writing and outcomes were impressive and exciting. Overall parameters were essential to the manuscript. This manuscript could be of interest to the audience. However, some points need to be addressed to improve the quality of the manuscript.

Major concerns.

Response to general comments:  Thank you for the feedback provided. We have carefully reviewed your comments and made the necessary revisions to address your major concerns in order to improve the quality of our manuscript. We appreciate your constructive input and the opportunity to enhance our work.

Point-by-point response to Comments and Suggestions for Authors

Comment 1. COVID-19 vaccination status: The COVID-19 vaccination is an important factor in substantially reducing the chance of disease severity. I suggest adding further information on COVID-19 vaccination;

Comment 1.1. Doses - you may be using a categorial data such as - 0, 1, 2, 3, 4, 5+ - never, partial-, full-, booster

Response 1.1. Thank you for this suggestion. The distribution of cases according to vaccination status is included in Figure 1 and was categorized as unvaccinated, complete, and incomplete vaccination schedules, according to the information available in the databases. We included additional details about the specific proportions according to the categories of vaccination status in the results section, page 11, lines 211 to 215.

Comment 1.2. Last dose interval to infection - suggest analysing further data to reveal insight information. As you know, recent vaccination within 3 months could protect the host from severe conditions than late - <14 days, 14-35 days (1 month), 2 months, 3 months, over 3 months.

Response 1.2. Thank you for this comment. The date of vaccination is available in the data sources from Brazil and Colombia, and we conducted a descriptive analysis to assess the completeness of this variable. Unfortunately, we identified significant issues with missing and incomplete dates in this variable; for this reason, we did not include this type of analysis. However, we have addressed the relevance of this issue in the discussion section, highlighting the importance of improving data accuracy and completeness in national surveillance databases. This was included in the discussion section, page 14, lines 322 to 328.

Comment 2. Did this study have smoking status data

Response 2. Thank you for the suggestion. Data regarding smoking status is available only in Mexico’s database. This variable was not recorded in the data sources from the other countries. We have revised Supplementary Tables 3-6 to include information about the smoking status of COVID-19 cases in Mexico for each outcome.

Round 2

Reviewer 2 Report

Comments and Suggestions for Authors

This version is fine.